# Soft Molds with Micro-Machined Internal Skeletons Improve Robustness of Flapping-Wing Robots

**DOI:** 10.3390/mi13091489

**Published:** 2022-09-07

**Authors:** Hang Gao, James Lynch, Nick Gravish

**Affiliations:** Department of Mechanical and Aerospace Engineering, Univeristy of California San Diego (UCSD), La Jolla, CA 92093, USA

**Keywords:** soft robot materials and design, elastic energy exchange, resonance, compliant components

## Abstract

Mobile millimeter and centimeter scale robots often use smart composite manufacturing (SCM) for the construction of body components and mechanisms. The fabrication of SCM mechanisms requires laser machining and laminating flexible, adhesive, and structural materials into small-scale hinges, transmissions, and, ultimately, wings or legs. However, a fundamental limitation of SCM components is the plastic deformation and failure of flexures. In this work, we demonstrate that encasing SCM components in a soft silicone mold dramatically improves the durability of SCM flexure hinges and provides robustness to SCM components. We demonstrate this advance in the design of a flapping-wing robot that uses an underactuated compliant transmission fabricated with an inner SCM skeleton and exterior silicone mold. The transmission design is optimized to achieve desired wingstroke requirements and to allow for independent motion of each wing. We validate these design choices in bench-top tests, measuring transmission compliance, kinematics, and fatigue. We integrate the transmission with laminate wings and two types of actuation, demonstrating elastic energy exchange and limited lift-off capabilities. Lastly, we tested collision mitigation through flapping-wing experiments that obstructed the motion of a wing. These experiments demonstrate that an underactuated compliant transmission can provide resilience and robustness to flapping-wing robots.

## 1. Introduction

In an effort to develop ever smaller high-performance flapping-wing micro-aerial vehicles (FWMAVs), roboticists have looked to flying insects for inspiration [1,2]. In the last decade, they have achieved major successes, including controlled tethered [3,4] and untethered [5,6,7] flight as well as the integration of sensors [8,9,10]. However, FWMAVs continue to be relatively delicate, made of lightweight, brittle materials in an effort to maximize strength while minimizing weight. If they are ever to be deployed in the unpredictable environments of the real world, micro flying robots will need to become more robust. As the scale of robots decreases, the space they are working in needs a higher ability to squeeze, stretch, and morph to adapt to a more complex environment [11].

The compliance of the insect flight anatomy is one factor that helps insects avoid damage that would otherwise cause catastrophic failure. Many insects and other arthropods have highly elastic proteins located in their tendons, wing joints, and in patches of their exoskeleton [12,13]. Most flying insects flap their wings by transmitting the force of powerful flight muscles through a deformable thorax, which is thought to reduce flight power requirements via the storage and release of elastic energy [14,15]. The functionality of elastic proteins is not limited to energy storage; however, it also plays a critical role in the flexibility and deformability of the exoskeleton and in the reduction of fatigue and damage in joints, wing veins, and other anatomical elements [13,16]. Additionally, elastic elements in the thorax are critical for coupling the left–right wing motion in a way that enables wing coordination, while also allowing for variation of wingbeat kinematics between left and right wings [17].

Inspired by the elastic elements in insects, and also driven by the engineering promise of resonant efficiency, roboticists have incorporated elastic elements into FWMAVs [18,19,20]. Elastic energy storage and return in the actuation system can improve flapping wing efficiency [21], and the addition of flexible passive wing hinges enables wings to generate lift on both the up- and down-strokes without requiring direct control of the wing pitch [22]. However, most attempts at incorporating elastic elements have maintained rigid linkages between wing and actuator, with a spring in parallel [23]. Thus, the kinematics of the wings are directly coupled to the kinematics of the actuator(s). This tends to simplify control, but it also means that any collisions are transmitted directly back to the actuator.

Previous flapping-wing robots have been designed to help with load balancing between wings and to mitigate wing collisions. Inspired by the wing flexibility in some wasp wings, roboticists incorporated a non-linear spring into the middle of the a flapping robot wing [24]. This work demonstrated that a non-linear stiffness hinge could enable large wing flapping amplitudes without buckling, but when the wing collided with an object, the hinge buckled and reduced the overall turning moment experienced by the robot. Similar work studying beetle wings and a beetle-inspired flapping robot demonstrated collision resilience through wing hinges [25]. Lastly, non-linear stiffness elements were incorporated into a quadrotor flying robot, which also displayed collision resilience due to body compliance [26]. Collision resilience can also arise due to actuator and transmission compliance, as demonstrated by a dielectric elastomer actuator (DEA)-driven flapping robot [27].

Underactuated mechanisms are commonly employed in other areas of robot design, such as robot hands and limbs; however, they have not been extensively employed for flapping-wing systems. A notable exception is in the design introduced by Sreetharan and Wood to achieve load-balancing between left and right wings. Sreetharan developed an underactuated transmission (analogous to a car differential) that could balance uneven torques from the left and right wing of a flapping robot [28]. This robot was able to accommodate asymmetric wing damage through the passive mechanics of the transmission.

We believe that incorporating compliance and underacatuation into the transmission design of a FWMAV may lead to improved resilience and flight performance. In the following work, we develop a compliant, underactuated transmission for flapping-wing robots. The goal of this transmission is to provide elastic energy exchange and robustness to collisions for a robot. We first present a new fabrication method for this transmission and the full four-winged robot (Figure 1). Next, we perform benchtop characterization experiments of the components, as well as lift-off and performance measurements of the robot. Lastly, we discuss the implications of this work and limitations still to overcome.

## 2. Robot Design

Our robot comprises four main elements: (1) a compliant thorax, (2) a set of four wings, (3) an actuator—either linear voice coil or rotary micro DC motor, and (4) a chassis and legs. An overview of these components and the fabrication methods employed can be seen in Figure 2. In the following section, we describe the design and fabrication of these components.

### 2.1. Transmission

#### 2.1.1. Kinematic Design

We used fmincon() in MATLAB with optimized parameters as $3<L_{1}<4.2$, $2<L_{2}<5$, $0<L_{3}<5$ and initial conditions set as [5, 8, 3] for $L_{1}$, $L_{2}$ and $L_{3}$. By optimizing, we derived the length as [4.2, 2.0, 0.88]; upon inspection with small difference, we used $L_{1,2,3}=\left[4.2,2.0,0.8\right]$ as the transmission kinematics coefficients.

To enable wing robustness and mitigate the effects of wing-structure collisions, we selected an underactuated transmission design that maps the single linear actuator across a parallel linkage to multiple wings. To further illustrate the potential of such an underactuate structure, we chose a four-wing configuration (Figure 3). The transmission kinematic design is inspired by the single-actuator Harvard microrobotic flying insect [29], which uses a symmetric dual-four-bar linkage system to generate wing motion. The actuator was attached to the central link of the transmission and drives the four linkages with a periodic force signal. The linkage lengths, $L_{1}$, $L_{2}$, and $L_{3}$, were optimized numerically to maximize wing sweep angle for the specified actuator stroke (Figure 3a), and the transmission was designed to be supported by a CF chassis (see Section 2.4 below).

To determine the link lengths for our transmission, we used numerical optimization to optimize for symmetric wing motion with maximum desired amplitude under the input actuation constraints.

#### 2.1.2. Fabrication

Initially, we designed the transmission out of silicone alone, but we found that the lack of rigid structure made driving the system challenging. However, when we clamped one tip of the transmission (Figure 4a), the whole silicone transmission deformed under the self weight of the structure. Furthermore, we examined this silicone transmission without an internal skeleton on the robot, actuated by voice coil. The initial stroke position was the neutral position, and the one quarter period and three quarter period, respectively, indicated two extreme positions. However, in the one quarter period (left extreme position), just the silicone transmission deformed (shown in Figure 4b) caused by the actuator input, which undermined the transmission stroke performance and wingstroke output. The solution was to embed a carbon fiber laminate skeleton into the molded silicone (Figure 5a). In fact, we note that adding silicone to CF+Kapton components has been done before (Rosen et. al. [30]); however, in that previous work, it was a very light coating, not a mold as we present here.

The fabrication of our compliant thorax required the combination of two different fabrication methods: small-scale, high-precision silicone molding for the elastic components and smart composite manufacturing (SCM) methods [31] for the creation of the rigid internal skeleton. We developed a hybrid fabrication process inspired by shape deposition manufacturing [32] that supports flexibility at wing joints and structural rigidity in the body elements.

The fabrication of our transmission included two steps: (1) we use the SCM method to form the “skeleton” of our transmission, and (2) we mold a protective silicone layer over the transmission (Figure 2a–f). The internal skeleton thus defines the approximate “rigid” kinematics of the transmission, while the protective silicone layer dictates the elasticity of the structure and provides a resilient, protective layer. In previous work, Zhou & Gravish [33] developed a process for centimeter-scale silicone structures that used milled machine wax templates for casting. In this work, we extended this process to incorporate the internal skeleton.

The skeleton was fabricated via a typical SCM process (Figure 2b–c), wherein 3D models of the laminated transmission were split into CF (0.2 mm thickness), heat-sensitive adhesive (Pyralux 1500), and kapton (25.4 micron, 100 HN) flexure layers and cut using a DPSS laser (DCH Laser, Photonics Industries International, Inc., Ronkonkoma, NY, USA). The layers were then aligned and heat pressed (50 psi, 350 °F, 30 min ramp up and down, 2 h hold) before a release cut freed the components of the skeleton. The components were then assembled into the final structure by hand and prepared to insert into the mold.

Silicone casting molds (Figure 5a) were fabricated out of machine wax (High-Speed Machining Wax, Bantam Tools) using a Micro Mill (Othermill, Bantam Tools Desktop PCB Milling Machine). The wall thickness of the silicone was set to 3.5 cm, which was determined via elasticity and blocked force requirements of the transmission discussed later in Figure 6. To determine the mold geometry, we computed the geometric “footprint” of the silicone transmission in Solidworks and then converted to G-code. The machining was performed with a 1/32${}^{\prime \prime }$ diameter flat end mill. The machining depth was constant and held at 3.5 cm, which provided a flat planar surface for the top and bottom of the transmission. To center the skeleton within the silicone mold, we added four carbon fiber tabs on each side as retainer plates on the wall of the mold to constrain the skeleton along the center line of the transmission. This ensured that the kapton joints of the internal skeleton aligned perfectly with the silicone compliant joints (Figure 5a,b).

We tested a range of silicone materials (Dragon Skin 10–60, Smooth-On). After a series of tests (See Results and Figure 6a), we ultimately found that Dragonskin 30 performed best for our purposes. We injected the silicone liquid into the mold using a syringe with a 0.8 cm diameter nozzle to fill the space between the CF skeleton and the mold, then degassed the silicone in a vacuum chamber with − 0.07 atmospheric pressure lasting for 20 min to remove air bubbles. We then allowed the silicone to cure for 16 h before removing and testing. The completed transmission was released from the mold using tweezers, taking care not to damage the silicone (Figure 2e–f).

### 2.2. Wings

#### 2.2.1. Wing Geometry

Insect wing shape, size, and structure vary widely [34]. We chose not to directly mimic any specific insect wing, and instead followed the wing shape of previously developed MAVs, including the Harvard RoboBee [3] and the UW Robofly [35]. The wing structure was composed of transparent thin polymer (5μ m thickness) supported by a CF frame with a thick leading edge and spars to support the trailing edge (Figure 7a), and it was fabricated using SCM lamination methods.

#### 2.2.2. Wing Hinge

Insects and hummingbirds pitch their wings so that they can produce lift on both the upstroke and downstroke [36]. We implemented passive CF laminate wing hinges to achieve this effect without requiring direct control of the wing pitch (Figure 7b–f). To generate higher lift, we designed the wing hinge with a mechanical stop as a pitch angle constraint. As shown in Figure 7d, we designed the gap of the hinge $B_{gap}=0.08$ mm based on the thickness of the carbon fiber sheet to make two sides of the carbon fiber part collide when the wing pitch reaches $\approx 50^{{}^{\circ}}$. In practice, the wing tends to hit the mechanical stop at 50° in one direction and 55° in the other (Figure 7g. The simulation work from Whitney & Wood [2] shows that the $45^{{}^{\circ}}\to 65^{{}^{\circ}}$ range angle-of-attack at mid-stroke will result in lower aerodynamic damping and better efficiency, assuming sinusoidal flapping and symmetric sinusoidal wing pitching).

### 2.3. Actuation

An important step for achieving flapping-wing flight is the selection of an actuator. A variety of actuation schemes, including piezoelectric (PZT) bending actuators [3,5], soft DEA actuators [22], small DC Motors [18,19], and electromagnetic coil actuators [37], have been used for small-scale flapping flight. PZT actuators and DEAs have relatively high power density (DEA has 300 W/kg and a lifetime of over 600,000 cycles [22]). However, PZT is fragile and DEA is hard to fabricate, and both require extremely high actuation voltages. A voice coil motor as an oscillating electromagnetic actuator can generate a linear motion with a relatively higher power density (15 kw/kg) [38], so we chose that. DC motors are cheap, well-developed, and have a linear relationship between current and force with fixed voltage. We focused on a design that incorporated a voice coil as our primary linear actuator. Then, aiming at lift force and universality, we tried to focus on the rotary motor. Thus, we tried two different actuators: a linear voice coil actuator and a micro rotary DC motor.

There were two additional important reasons for selecting a rotary actuator. The first, although linear motions controlled via a DC motor and slider-crank assembly have a fixed amplitude, our silicone transmission can still amplify the amplitude of the wing flapping angle at resonance by soft silicone transmission joints, even with a fixed input slider stroke. Second, even though the slider-crank has excess longitudinal component force on a slider when we apply the slider on horizontal motion, we can solve the longitudinal component force by setting an offset position for the slider, and the silicone transmission itself can also tolerate some of the longitudinal component force.

We used a small voice coil motor (LVCM-010-013-01, Moticont). The weight and the scale of the motor was relatively low, which is critical, as it makes up the bulk of the overall mass of the robot (63.3%). The magnetic core was the only moving part of the linear actuator, and it was fixed to the carbon fiber cross-bar on the transmission. The core had a lower mass than the wire coil assembly, so it generated relatively lower inertial effects as it moved back and forth. The rated stroke of the motor was 6.4 mm, although in practice, the stroke was closer to 3.3 mm after incorporating it with the transmission and chassis using a high temperature tolerance paste(J-B Weld 37901).

For the DC motor case, we utilized a micro DC motor, (136:1 Sub-Micro Plastic Planetary Gearmotor). The stall torque was 550 g/cm at 6 V. The weight made up the bulk of the overall mass of the robot (41.67%). As shown in Figure 8, to decrease the friction on the slider crank, the rotary motor transferred the torque to a Slippery PTFE crank (9266K83, McMASTER-CARR) of the slider crank. We choose a formable brass rod (8859K481, McMASTER-CARR) as the connecting rod and set the revolute joint (Slippery Delrin material, 8578K411, McMASTER-CARR) onto the carbon fiber cross-bar on the transmission as a slider to make the input motion move back and forth. In practice, the stroke was 5.2 mm totally. Other than that, to maintain the brass rod connected with the crank in-plane, we used soldering on the tip of the rod as a shoulder of the rod, as shown in the Figure 8 zoom-in, which is the bottom view of the robot prototype.

### 2.4. Chassis and Structure

The chassis forms the primary structure of the robot. It has two purposes: (1) support the actuator and transmission and (2) provide attachment points for legs and guide rail support structures during vertical lift experiments.

Initially, we used a 3D printer to create the motor chassis. However, that version had two disadvantages: thermal tolerance and mass density. The heat (over 150  °C after running the voice coil for 10 s with 3 A current) produced by the voice coil motor can quickly melt the PLA material structure. Therefore, we designed the motor chassis out of carbon fiber, like the transmission’s internal skeleton. The motor chassis was composed of a carbon fiber sheet (0.25 mm thickness) assembled via slot joints and fixed using cyanoacrylate glue (Loctite 495) to create a rigid structure (Figure 2i).

### 2.5. Assembly

Once all individual components were fabricated, we assembled the robot manually. We first attached the transmission to the chassis (Figure 2f,i) using alignment tabs and securing with cyanoacrylate (CA) glue. Once the transmission was secured, we inserted the actuator. Then, for the voice coil motor case, we glued the magnetic core to the transmission using CA glue, which was held within the chassis by a press fit design. For the rotary motor case, we glued the revolute joint (Slippery Delrin material, 8578K411, McMASTER-CARR) part onto the carbon fiber cross-bar on the transmission. Furthermore, we plugged the brass rod as the slider-crank connecting rod to the Delrin revolute joint part and the PTEE crank. As shown in Figure 5b, in the cross-bar of the silicone transmission, there is a carbon fiber circle holder for the magnet core. This circle has the same diameter as the magnet core, and we used the high heat tolerance paste (J-B Weld 37901) to align the core and transmission together as concentric circles. Then, we clamped the motor coil to the motor holder circle of the motor chassis with the first motor holder circle along with the motor edge to make sure that we could set the magnet core in the neutral position (Figure 2h,i). For the rotary motor case, we computed the length of the brass connecting rod by setting the silicone transmission in the neutral position in SOLIDWORKS to make sure the slider-crank moved the same distance for back and forth motion. We adjusted the small offset distance of the slider by exciting the silicone transmission motion with power supply until the four wings flapped with the same angle. Last, we used soldering to add a shaft shoulder on the tip of the connecting rod to limit the motion between the connecting rod and crank.

## 3. Results

### 3.1. Transmission Kinematic and Dynamic Characterization

In our first experiments, we sought to characterize the force and kinematic behavior of our transmission. We constructed transmissions using silicone of different stiffnesses (Dragon Skin 30 and 60, Smooth-On) and measured the force it takes to deflect the transmission under compression and extension. We also measured the kinematic relationship between input displacement and output wing motion.

We mounted the transmission into the motor chassis and fixed it to a linear XYZ stage (Thorlabs, M-DS40). We mounted a strain gauge (Eujgoov, 0–100 g, 0.010–0.020% accuracy) to a motorized translation stage (Thorlabs, MTS50-ZB) with 3D printed gripper and attached the other end to the cross bar motor connection of the transmission. Starting in the neutral position, we moved the motorized stage, measuring displacement and force from the strain gauge (NI USB-6003, 1 kHz sample rate).

Figure 6a shows the results of the experiment using different versions of the transmission. Ideally, we want to transmit 100% of the motor force to the wings, so we want to minimize the force it takes to induce a certain displacement because of the actuators’ block force limitation. The green region in the figure indicates the case where it takes less than 200 mN to deflect the cross bar.

We tested five different transmission designs using one of two types of silicone, Dragon Skin 30 or 60 (“Sil”). One design was 100% silicone (no skeleton), and the rest had some kind of CF skeleton (“Sk”). For the designs that had a skeleton, some had skeleton going out to the wing connections (Tip = Y) and some had a skeleton only at the cross bar motor connection (Tip = N). Finally, we had two thicknesses for the compliant silicone joints; “Hi” indicates a thick joint (>1 mm) and “Lo” indicates a thinner joint (<0.9 mm) (see Figure 3b and Figure 5b).

We found that lowering silicone joint stiffness improves the performance of the transmission, as does incorporating a rigid skeleton. We also found out that while Dragon Skin 60 compressed very easily, it was more viscous than Dragon Skin 30, which could hinder elastic energy exchange. Moreover, in the rest of the experiments, we concluded that including only a rigid cross bar (Tip = N) leads the silicone transmission to be deeply stretched when driving at high frequencies, which further decreases the magnitude of flapping. The final configuration was therefore constructed with Dragon Skin 30, a full skeleton, and thin silicone joints.

We optimized the kinematics of the silicone body length with $L_{1}$, $L_{2}$, $L_{3}$ as [4.2, 2.0, 0.8] mm and then derived the linear relationship in full stroke between the actuator displacement and magnitude of wing flapping. By using the Moticont voice coil motor (LVCM-010-013-01), with the actuator full stroke displacement from −1.8 mm to 1.5 mm (compression motion is negative and stretch motion is positive), as desired, the transmission generated analog linear motion to wing rotation. Moreover, the blue line is the transmission kinematics at statics from the experiment. Besides, the designed full stroke data was 6.4 mm; however, because of the re-design of the motor and assembly, our full stroke at statics was 3.3 mm as shown in Figure 6b. To derive the experimental data for flapping-wing magnitude sweeping with motor displacement, we glued (glue gun) the feet of the flapping robot on the table and put a ruler into the frame of our high speed camera (Phantom-VEO-L) with 2000 fps. After exciting the actuator with a pretty low velocity, we acquired the blue data line in Figure 6b.

### 3.2. Transmission Fatigue

Transmission integrity is crucial to FWMAVs that play a distinct role in lifespan. One critical challenge facing the FWMAVs transmission is load stress on the hinge, which will significantly influence the lifespan of the transmission hinge.

The CF + Kapton components can increase length and width, substantially reducing the stress in the hinge, for which stress is contributed significantly by the off-axis loads (Malka et al. [39]). However, for a larger scale of FWMAVs with a CF+Kapton components transmission, a hinge that is too long and wide will crimp the transmission hinges when underactuated, further undermining flapping precision and performance.

We tested three different transmission hinges: silicone with CF (with pin) + Kapton components, CF (with pin) + Kapton components, and CF (without pin) + Kapton components. These three sets of experiments swept off-axis angle $\phi_{off}$ from 0° to 10° within step at 5°. As shown in Figure 5a, the pin (retainer plate) is used for ensuring the silicone joints’ position along with the CF + Kapton components joints’ position. To simplify the fatigue experiment, instead of testing the whole transmission with a full wing load, we examined the single transmission hinge with a paddle. The motor generated around 3.3 Hz for one period rotary motion from paddle to the transmission hinges and one period within two clockwise rotation cycles and counter-clockwise cycles; this frequency was relative in corresponding to the frequency of the maximum lift force shown in Figure 9. We used a 12PPR resolution encoder (MGT YC2010-24, AB Phase Encoder) to read fatigue cycles, and we measured if the transmission hinges were broken to define hinge failure.

The results (Figure 10c) show that the transmission hinge of silicone + CF (with pin) had a distinctly longer lifespan than the CF (with and without pin) from 0° to 5° off-axis stress. However, once the paddle hit the transmission tips (Figure 10a), the pin in this silicone + CF transmission had also been touched and generated higher friction, resulting in higher centrifugal pull. This is the reason that the transmission hinge of CF with pin has (purple line) a much lower lifespan than the CF without pin. Additionally, for off-axis at 10°, the silicone + CF (with pin) had a higher contact area and higher surface friction than the CF (without pin); thus, the silicone + CF performance was similar with the CF one.

### 3.3. Elastic Energy Exchange of the Robot Wingbeat

For effective elastic energy exchange, the system should be operated at the resonant frequency of the spring transmission and wing inertia [1]. For the voice coil motor case, we fixed the feet of one of our flapping robots on a scale (PMW-320, Intelligent Weighing Technology Laboratory Balance). Driving the voltage at 31V, we found out that the maximum magnitude of flapping-wings was corresponding with the maximum lift force, as shown in Figure 9, and the error bar is the standard deviation value in the data set at each frequency. Then, we went further and deeper to test the rotary motor case. After, we fixed the robot body onto the force sensor (as shown in the Figure 11a), and we found that the maximum lift force needed higher frequency than the need of the maximum flapping amplitude. Usually, the lift force peak and the amplitude resonance do not coincide. This phenomena was introduced and studied by Zhang and Deng [40].

Furthermore, there are multiple “resonances” associated with different kinematic and dynamic variables. The kinematic resonance is where wingbeat amplitude is maximized as a function of frequency, ω. The lift resonance is where mean lift force is maximized as a function of frequency. These two resonance often do not coincide, and instead, the lift resonance is always at a higher frequency than the kinematics resonance.

Moreover, based on the rotary motor test, the frequency was 5/4 voltage. Thus, the higher maximum lift force could be generated at around 12 V rather than 31 V from the voice coil case. We can conclude that the rotary motor performance was better at lift force and flapping amplitude than the voice coil motor case.

Besides, take Figure 6a for example—the higher thickness of the compliant joints on the silicone transmission not only refers to a higher blocked force, but it also had a lower resonance frequency with higher damper performance. Thus, to have a better efficiency on elastic energy exchange, we followed the idea of keeping our thickness relatively low.

### 3.4. Free Lift-Off

We next attempted to achieve take-off from the ground and air. We placed the robot on the brass rails (shown in Figure 11) and video recorded with a high-speed camera (Phantom-VEO-L) at 2000 frames per second. The actuator was driven at a voltage from 8 V to 11 V, meaning the frequency was excited roughly from 10 Hz to 13.75 Hz. In Figure 12, we show an example of take-off from a robot on vertical rails to only allow vertical motion, and in the Appendix A, we also show another take-off experiment from a hanging robot (i.e., no leg bouncing effects). Since the robot was not under closed-loop control flight, it was very unstable, and in all take-off attempts, the robot would quickly roll or pitch. However, free take-off experiments consistently demonstrated the robustness of the robot design. The compliance of the transmission was able to absorb the actuator motion and kept the wings from sustaining too much damage. This prompted us to explore how the transmission can act as an underactuated mechanism to distribute load between wings that may become blocked by obstructions.

### 3.5. Robustness of the Robot Wingbeat

The rigid kinematics transmission always stopped moving when collision happened. Especially for four wings with rigid transmission, if they blocked one single wing movement, the rest of the three would stop moving. However, when we tested our soft silicone transmission with an internal skeleton, we found that the rest of the three wings would still move even after we locked one wing.

Then, we tried to go deeper to inspect the lift force (by using the Futek load cell MODEL LSB200, which is used for lifted force tests, and we drove this one degree of freedom load cell by device USB-6003, with a 1000 sample rate and 100 number of samples.) and flapping magnitude of wings when we locked one single wing. When we barely attached a one-dimensional force sensor onto the bottom of our flapping robot with a signal amplifier, we found the force data was always ruined with the inertia force from the magnet core motion (front and back motion). Thus, we set a sliding rail on a motor to the front and back side to eliminate the magnet inertia force. Initially, we used a carbon fiber rod with a carbon fiber circle gripper. However, the contact friction between carbon and carbon was high, and the circle gripper on the two sliding rails would generate a big overturning moment. Thus, we utilized a kind of brass sliding rail (Ultra-Machinable 360 Brass Rod, 3/32${}^{\prime \prime }$ diameter) to lower the contact friction and designed two flat plate grippers orthogonally arranged on each sliding rail. Even with tiny fabrication and assemble errors on the motor chassis and sliding rail on an acrylic(0.138${}^{\prime \prime }$ thickness) plate, these orthogonal flat plate grippers would tolerate them. Besides, we designed a wing locker as tips on the slider to clamp the wing frame from two sides and ensure there was no force going to the downside from the locker to the wing and, finally, to the force sensor.

Shown in Figure 11e is the mean lift force in one period with a 12 Hz frequency, which is the maximum flapping amplitude in Figure 9. In Figure 11f, with the locked Wing 1, Wings 2–4 show a different performance in flapping magnitude.

## 4. Discussion

We have developed a flapping-wing robot with a compliant transmission, fabricated using a hybrid SCM and silicone molding technique, driven by a voice coil linear motor. We performed a range of benchtop experiments to characterize each of the components of the system. The full system had a characteristic resonant frequency that maximized the wingbeat amplitude and lift force, and it was able to achieve lift-off for brief periods of time. The robot, like other FWMAVs, was unstable—it could not maintain open-loop flight—but it also had the unique feature of being able to continue to generate wing motion and lift even if one wing was restricted, thanks to the compliant transmission.

One major challenge to achieving sustained flight in this robot was that we required high power input even to achieve brief lift-off. We found that we could generate significant lift but only if we drove the motor at voltages that would cause it to burn out within a matter of seconds. This was primarily due to the inefficiencies in voice coil actuation [41]. Ultimately, further actuator development may be necessary to achieve successful implementation of this type of compliant transmission in an FWMAV.

The novel fabrication techniques we have used are part of a wider trend towards robustness and resilience in mobile robots. Robots such as the 16g DASH robot [42] leverage lightweight materials and actuators to achieve high-speed running and resistance to damage from collisions and falls. Others use soft-robotic and compliant structures to build robots that are resistant to crushing and can navigate tight spaces [43,44,45]. Other researchers have incorporated bio-inspired collapsible wing features into flapping robots [24] that are able to dampen collisions with obstacles. Our work extends the principles inherent to robust robotic design and applies it to the transmission of flapping-wing robots, where it has the potential to provide a new level of resilience to future small-scale flying robots.

## 5. Conclusions

Flapping-wing robots have been a critical focus in aerial robotics over the last fifteen years, and the pace of development has been rapid. New actuators, power autonomy, and control capabilities have been demonstrated in flapping-wing robots. However, unlike their biological counterparts (flying insects, for example), flapping-wing robots can suffer from a lack of robustness due to the materials used and the rapid motions of the wings. In this work, we develop a novel compliant transmission for flapping-wing aerial vehicles that provides elastic energy exchange and robustness to flapping-wing robots. We demonstrate elastic energy exchange and resiliency to wing collisions in this robot in the experiment. These advances present new design opportunities for flapping-wing robots that may soon have to operate in crowded and obstacle-laden aerial environments.

## Figures and Tables

**Figure 1 micromachines-13-01489-f001:**
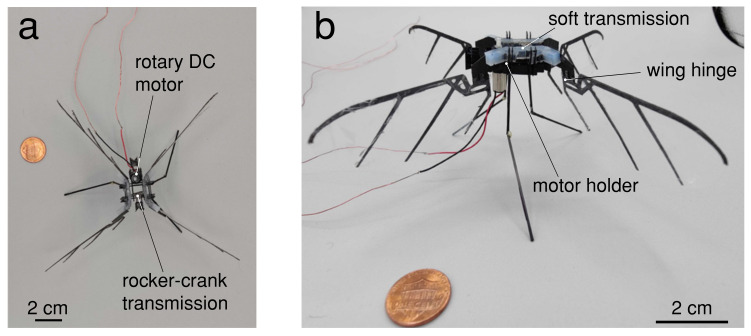
A 3.6 g flapping hummingbird-scale robot with soft transmission. The wing span is 21 cm, and the overall dimensions of the robot are 16 cm × 16 cm when the transmission is in the neutral position. (**a**) Top view, (**b**) front view.

**Figure 2 micromachines-13-01489-f002:**
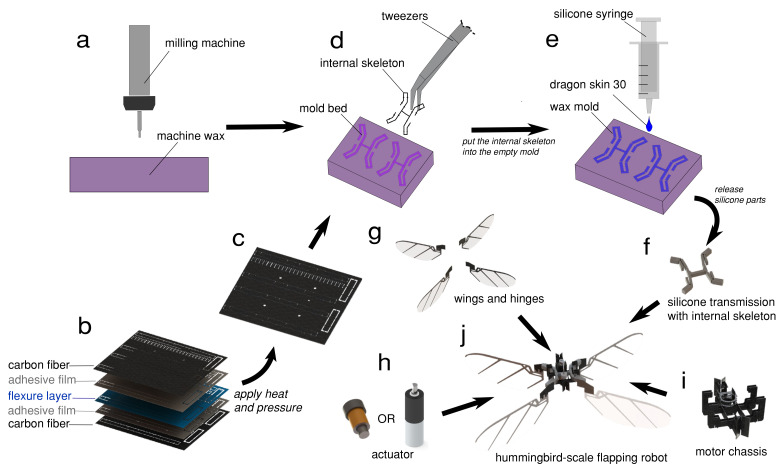
Overview of the manufacturing process. (**a**) The wax mold for the compliant transmission was machined using a 1/32${}^{\prime \prime }$ flat end mill in an Othermill micro-milling machine. (**b**) The skeleton of the transmission is a laminate constructed from a 5-layer stack of CF, adhesive, and thin, flexible polymer. (**c**) We apply heat and pressure to fuse the laminate, pop out the skeleton, and fold it into the proper shape. (**d**) The CF internal skeleton was placed into the mold bed, held in place by small horizontal tabs to prevent misalignment. (**e**) Mold was filled with silicone (Dragon Skin 30, Smooth-On), using a syringe to ensure an even fill. After setting completely, the compliant transmission (**f**) was assembled with the wings (**g**), actuator (**h**), and motor chassis (**i**) into the completed FWMAV (**j**).

**Figure 3 micromachines-13-01489-f003:**
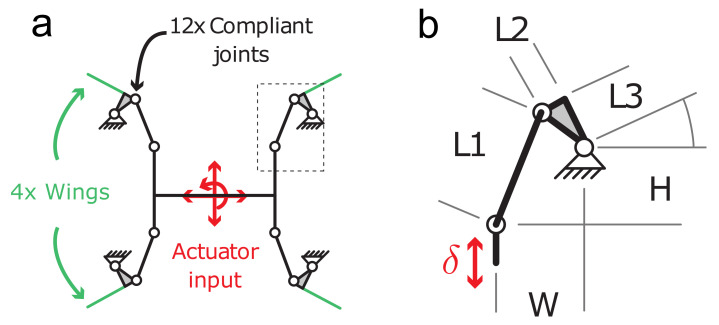
Kinematics and assembly of a compliant “thorax”. (**a**) Quad-four-bar linkage system that transmits linear actuation to rotational wing motion. White circles indicate flexure joints, green lines indicate wing locations, and the linkage is fixed to the chassis on the sides. (**b**) Details of the linkage, $L_{1}=4.2$ mm, $L_{2}=2.0$ mm, $L_{3}=0.8$ mm, W=3.0 mm, H=3.13 mm.

**Figure 4 micromachines-13-01489-f004:**
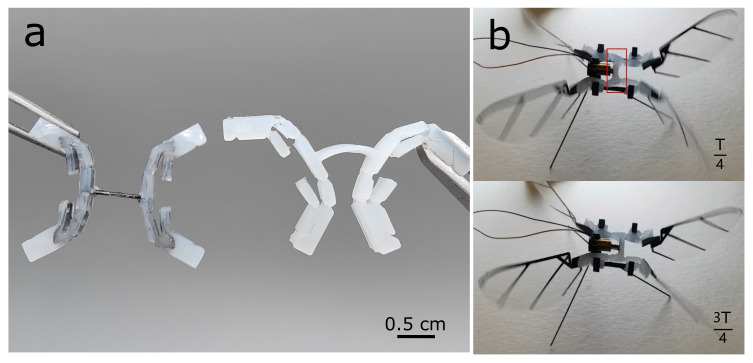
Comparison between a silicone transmission with internal skeleton (**a**), and a transmission made of just silicone (**b**). Silicone transmission deforms under the self-weight of the structure. (**b**) Example of dynamic deformation of silicone transmission without internal skeleton. The red box in the top image shows the actuator causing the transmission input to deform and undermine the stroke performance.

**Figure 5 micromachines-13-01489-f005:**
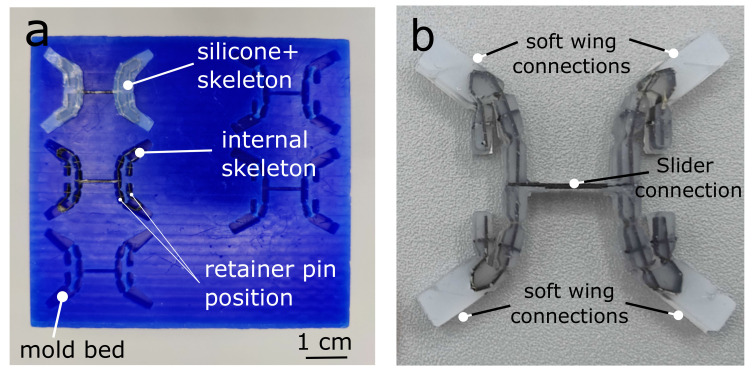
Assembly of a compliant “thorax”. (**a**) Photo of the machined wax mold. (**b**) Completed transmission with carbon fiber skeleton and silicone mold.

**Figure 6 micromachines-13-01489-f006:**
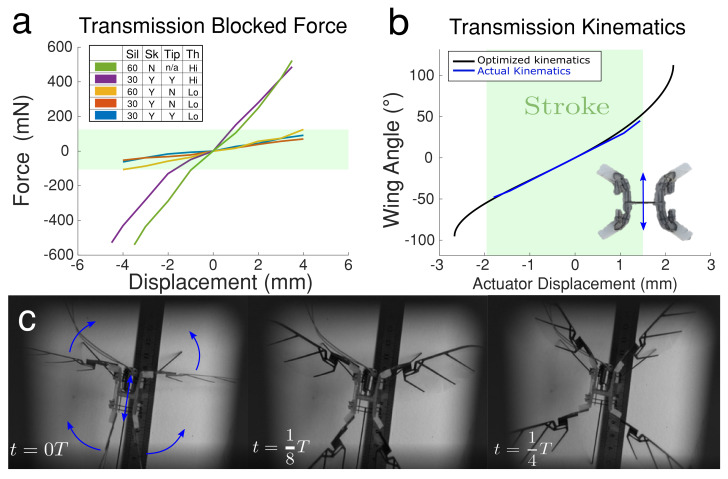
Dynamics and kinematics of the silicone transmission. (**a**) Blocked force necessary to induce displacements for different designs of the transmission (see text). (**b**) Transmission kinematics, with a linear relationship between actuator displacement and flapping magnitude (blue) and optimized curve (black). (**c**) Video stills show wing motion over 1/4 of a period, *T*, using the linear voice coil actuator.

**Figure 7 micromachines-13-01489-f007:**
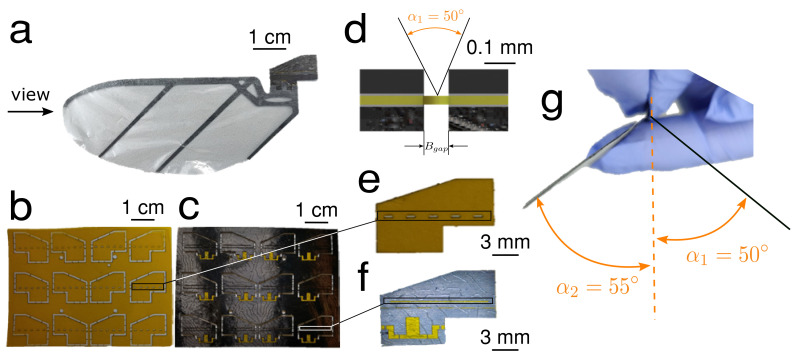
Robot wing and hinge design. (**a**) Side view of wing structure with wing frame and passive hinge. The hinge is constructed as a laminate with flexure (**b**) and CF layers (**c**) laminated with adhesive layers (Pyralux 1500). (**d**) Cross-sectional view of wing hinge, indicating mechanical stop angle. (**e**) New Kapton membrane was cut off a raster-like space to decrease the stiffness of the hinge structure to increase the acceleration of the angle-of-attack when flapping. (**f**) We added a groove to the wing to wick any glue spillage to protect the wing hinge gap from interference from the glue. (**g**) The mechanical stop ensures that the angle-of-attack at two mid-strokes is near 50° in both directions.

**Figure 8 micromachines-13-01489-f008:**
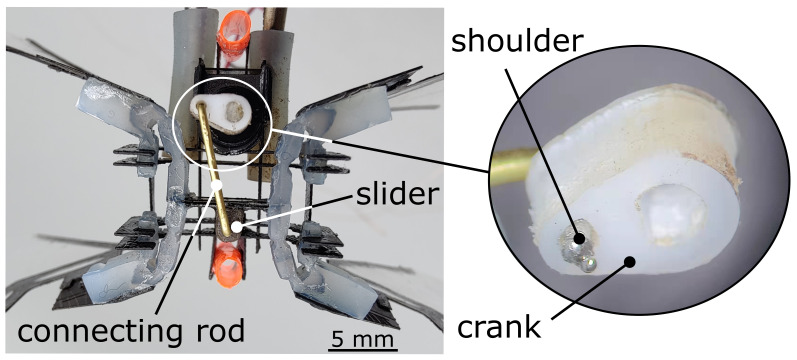
Close-up view of the slider crank; the soldering tip on the connecting rod serves as a rod shoulder to limit the crank in-plane.

**Figure 9 micromachines-13-01489-f009:**
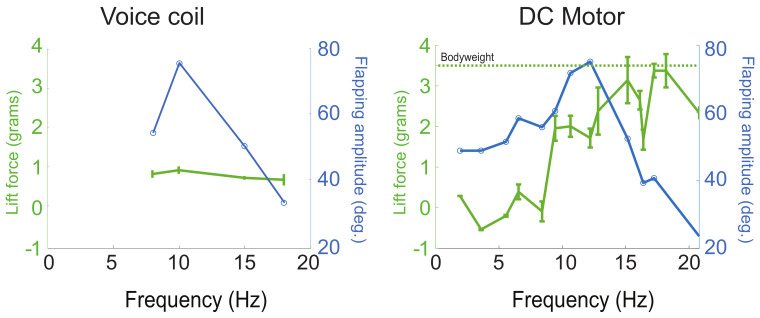
Lift force and flapping amplitude vs. frequency for the voice coil-driven robot and the rotary motor-driven robot. Both show a resonant peak characteristic of elastic energy exchange, but the rotary motor produces significantly higher lift. The dashed green line indicates the weight support lift threshold.

**Figure 10 micromachines-13-01489-f010:**
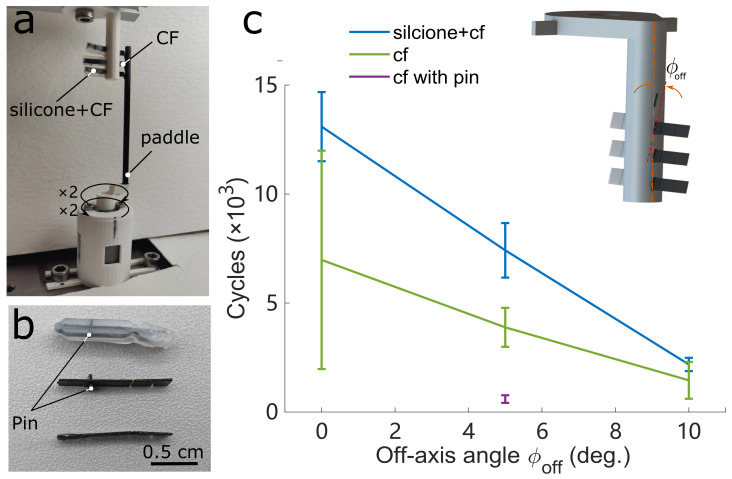
Comparison of the fatigue resistance between different transmission hinge. (**a**) Experiment setup for testing the fatigue of different transmission hinges. (**b**) Different transmission hinges. Silicone + CF (with pin), CF (with pin) and CF (without pin) transmission hinges. (**c**) The blue line indicates the transmission hinge with silicone + CF has a higher life cycle period than the CF hinge and much higher than the CF with pin (which is the same pin on the transmission hinge with silicone + CF).

**Figure 11 micromachines-13-01489-f011:**
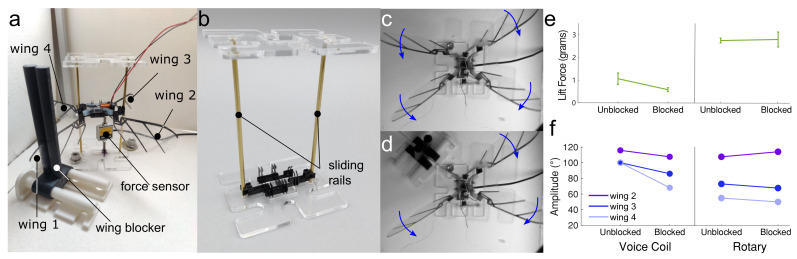
Overview of robustness test. (**a**) Detail of measurement setup with wings and wing blocker labeled. (**b**) The brass sliding rails on either side of the chassis restrict motion to vertical, reducing anomalous force sensor measurements. The underactuated design of the transmission means that flapping is possible even when one wing is restricted (**c**,**d**). (**e**) Lift can be produced even when a wing is blocked, although the rotary motor performed better than the voice coil motor. (**f**) Both voice coil and rotary designs maintained significant wingbeat amplitudes, but fabrication inconsistencies made it difficult to achieve symmetrical wingstrokes.

**Figure 12 micromachines-13-01489-f012:**
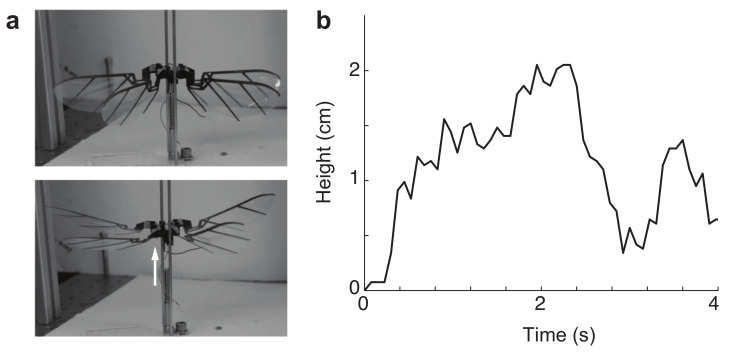
(**a**) Two frames from the Appendix A, showing the robot’s lift-off on a tethered rail. (**b**) Height versus time during flapping.

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
