# Peer review of "Soft Molds with Micro-Machined Internal Skeletons Improve Robustness of Flapping-Wing Robots"

_micromachines, 2022, doi:10.3390/mi13091489_

Round 1

Reviewer 1 Report

In this manuscript, the authors develop a compliant transmission for flapping wing aerial vehicles that provides elastic energy exchange and robustness to flapping wing robots. The authors optimized the structural parameters, validated design choices, and tested collision mitigation. Below are some comments/suggestions for the authors to consider to improve the manuscript.

1. The silicone embedded with carbon fiber improves the rigid of part and maintains its structure. I do not think the method to improve robustness of flapping wing robots is new. Meanwhile, silicone, as a soft material, has many uncontrollable degrees of freedom, which will lower the flying stability of flapping wing robots.

2. It is better to compare the flapping wing robots with others in previous literatures. Thus, the advantages of robustness of flapping wing robots can be obviously highlighted.

3. The tense is confusing in the text. Especially, it should be past tense for the experimental part, but the authors used the present tense. Meanwhile, the manuscript is suggested to write in the order of Figures, which will let the readers read it more smoothly.

Reviewer 2 Report

In this paper, the authors extensively presented the fabrication process of a millimeter flapping wing robot with the transmission soft mechanisms designed with or without internal skeletons for performance improvement. The authors did an excellent job in explaining the design and fabrication process. Although the resultant robot can not maintain open-loop flying, the presentation of the idea is clear and straightforward. The most concerning matter is the introduction of the skeleton, which seems to be intuitive by nature from a design perspective, as it would be strange to use silicone directly to design the transmission mechanism. The authors should take more care with the language and formatting of this paper, especially towards the end of this paper. I'm not quite sure about the innovation or contribution behind this paper and how the method presented in this paper is different from the method presented in previous work. I have no further questions regarding this paper. The work is publishable in the current form, after revising the language and formatting issues with care.

Reviewer 3 Report

In this paper, Gao et al. developed a compliant internal skeleton to improve the robustness of flapping wing robots. This paper demonstrates clear novelty and contribution to the field of flapping wing robotics. Overall, this paper is well-written and all the figures are clearly illustrated. The reviewer recommends for publication in the journal. The reviewer has a minor comment: how are the silicone materials determined? The authors used Dragon Skin 10-60 and found that Dragon Skin 30 is the best. What are the selection criteria (damping, elasticity)? The authors are recommended to test other silicone materials apart from the Smooth-on in their future works. Perhaps some silicone materials from Wacker may perform better than Dragon Skin.
